# CHALLENGING THE FOUNDATIONS: MINING HARD TEST SAMPLES THROUGH DIFFUSION GENERATION

## ABSTRACT

Large foundation models have achieved tremendous success with impressive performance in multiple applications. However, their performance is often benchmarked on natural images, where novel combinations of specific objects and nuisances can be missing and not tested. In this work, we develop a framework to efficiently probe foundation models for their vulnerabilities with diffusion generation, termed DiffusionExplorer. We show that our framework can efficiently construct a test set with novel combinations of object and nuisance factors to expose the failures of foundation models. Experimental results show that our mined test samples are challenging to foundation models, such as MiniGPT-4 and LLaVa, significantly reducing their accuracy by 29.56% and 39.96%, respectively. Our work suggests that generative models can be viewed as an effective data source in finding the vulnerability of large vision foundation models.

## 1 INTRODUCTION

Large foundation models (Bommasani et al., 2021; Li et al., 2022; OpenAI, 2022; Ouyang et al., 2022; Touvron et al., 2023) have made remarkable strides in numerous applications such as text-to-image synthesis (Rombach et al., 2022; Saharia et al., 2022; Ramesh et al., 2022; Ruiz et al., 2023) and visual question-answering (Zhu et al., 2023a; Liu et al., 2023; Li et al., 2023a). GPT-4 (OpenAI, 2023) stands out for its advanced conversational capabilities and exceptional multi-modality skills. However, those foundation models can fail under unforeseen user input (Wang et al., 2023; Zhao et al., 2023). Due to their wide adaptation, it is becoming increasingly important to identify and understand the vulnerabilities in large vision foundation models (Ming et al., 2022; Li et al., 2023b).

Prior work studied how to collect hard test samples for vision models. ImageNet-A (Hendrycks et al., 2021) filters out samples from online image collections, while Stylized-ImageNet (Geirhos et al., 2018) alters the styles of ImageNet images (Deng et al., 2009). Nevertheless, they cannot create images with unseen object and background combinations. Another way is to source images with the specified object and nuisances online. For instance, a search for `a plate on the ice skating rink outdoor` should return pictures of a plate on ice. However, as illustrated in Figure 1, Google could not return the expected image because this novel combination does not exist online. ObjectNet (Barbu et al., 2019) controls the backgrounds by manually collecting images in 4 different scenes. However, to achieve this, ObjectNet (Barbu et al., 2019) requires 5982 workers to collect images at their homes, which is time-consuming and labor-intensive.

We introduce DiffusionExplorer, a framework that is able to identify challenging test images for large foundation models using diffusion generation (Sohl-Dickstein et al., 2015; Ho et al., 2020). Leveraging diffusion models to generate images with novel combinations, DiffusionExplorer mines a hard test set by finding the failures across various surrogate models, where the surrogate models' predicted categories differ from the object category in the diffusion model's input prompt.

A key advantage of our approach is that it can create hard images with novel combinations that are absent from the existing image datasets and data source, see Figure 1. Since we can automatically generate and find the hard images, our test set can be scaled efficiently. Our approach shows that diffusion models can be an effective test set for exposing the vulnerabilities of large foundation models. Moreover, we envision our method will be more effective with future generative models that produce superior images.

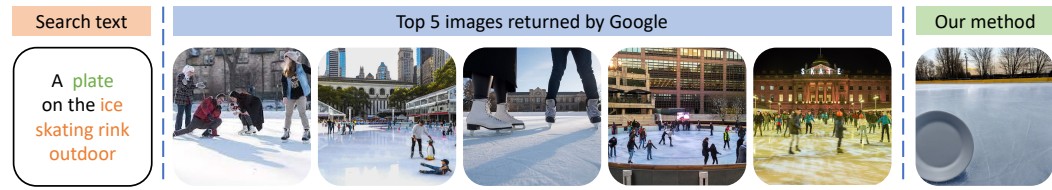

Figure 1: Top 5 images returned by Google when searching for a new combination of an object and a nuisance. We show that Google often fails to return images with the novel object and nuisance combination, indicating that Internet-image-based testing for foundation models can miss many corner cases. Our method can create images with novel combinations to test large foundation models.

Experiments show that our test set significantly reduces the test accuracy of new foundation models, suggesting that our test set poses a general challenge for large foundation models. For MiniGPT-4 (Zhu et al., 2023a) and LLaVa (Liu et al., 2023), our test set reduces the accuracy by 29.56% and 39.96%, respectively. Our test set also reveals the spurious bias(Hendrycks et al., 2021) of large foundation models and their failures in nearest neighbor retrieval. We find that our test set is also challenging to the latest GPT-4 model and can cause it to mispredict. The main contributions of this paper are summarized as follows.

- This work proposes the first framework that mines hard test images of large foundation models through diffusion generation, suggesting generative models can be a effective data source for testing foundation models.
- This work constructs a test set with novel combinations of objects and nuisances, which can be a useful benchmark to challenge future foundation models.
- Our test set significantly reduces the performance of large foundation models, e.g., an accuracy decrease of 29.56% and 39.96% for MiniGPT-4 and LLaVa, respectively.

## 2 RELATED WORK

**Large foundation models.** Large foundation models (Bommasani et al., 2021) have achieved significant strides in various applications (Raffel et al., 2020; Scao et al., 2022; Hoffmann et al., 2022; Smith et al., 2022; OpenAI, 2022), among which BERT (Devlin et al., 2018) and GPT-2(Radford et al., 2019) are two pioneering large language models that have inspired numerous advancements (Chowdhery et al., 2022; Ouyang et al., 2022; Touvron et al., 2023). Specifically, GPT-4 (OpenAI, 2023) is distinguished by its advanced conversational skills and exceptional multi-modality capabilities. Two followups LLaVa (Liu et al., 2023) and MiniGPT-4 (Zhu et al., 2023a) exhibit many capabilities similar to GPT-4 by visual instruction tuning and adopting a large language model Vicuna (Chiang et al., 2023), respectively. Despite the remarkable performance of large foundation models (Sun et al., 2023), they may fail under unexpected user input (Wang et al., 2023; Zhao et al., 2023). In this work, we challenge large foundation models using diffusion generation to reveal their vulnerabilities.

**Expose the vulnerabilities of large foundation models.** Previous research has introduced multiple benchmarks for evaluating large foundation models (Bai et al., 2023; Bitton et al., 2022; Xu et al., 2023; Wang et al., 2023). For out-of-distribution (OOD) capabilities, Vlue (Zhou et al., 2022) evaluates these models using images from MaRVL dataset (Liu et al., 2021). There are also other test sets available. ImageNet-A (Hendrycks et al., 2021) filters out easy images from the data online, while Stylized-ImageNet (Geirhos et al., 2018) modifies the styles of ImageNet images (Deng et al., 2009). However, they cannot obtain images with novel combinations of objects and nuisances, such as backgrounds. ObjectNet (Barbu et al., 2019) includes images of 4 backgrounds, which are collected by 5982 workers from their homes. This process requires high time and labor costs. In this paper, we show that our framework can efficiently find hard test images with novel combinations that do not even exist online, resulting in a significant accuracy drop of large foundation models.

**Large foundation models and diffusion generation.** Diffusion models (Sohl-Dickstein et al., 2015; Ho et al., 2020; Dhariwal & Nichol, 2021; Ho & Salimans, 2022) have demonstrated great success

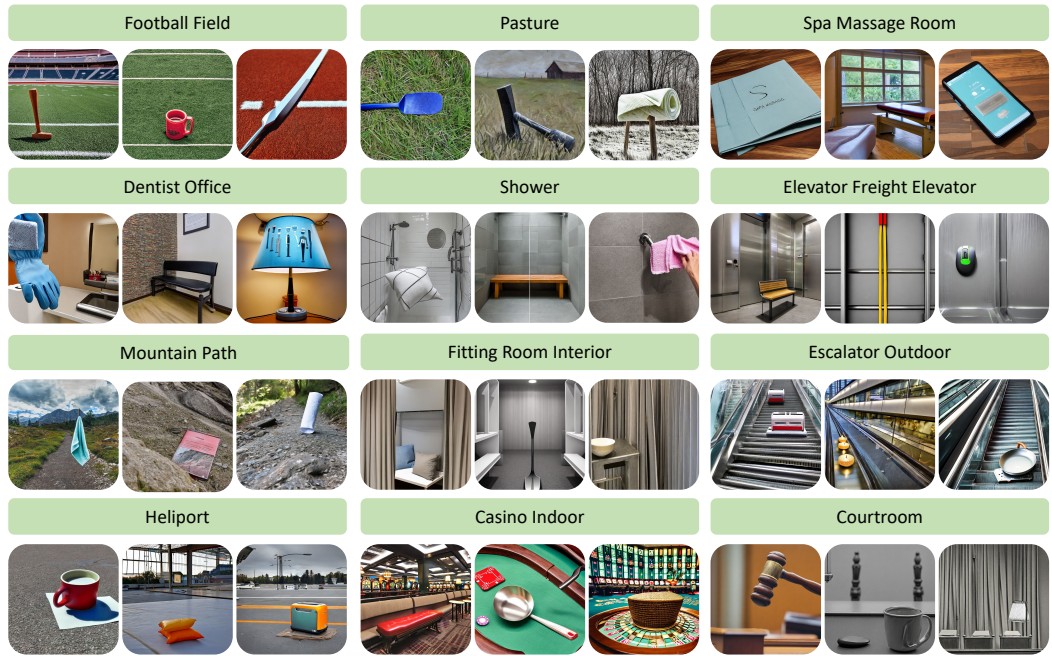

Figure 2: Examples of images with novel combinations in our test set, where each image corresponds to one object category and one background. We show 12 random backgrounds, and for each background, we show 3 random object categories. Our method, DiffusionExplorer, is able to construct a test set with novel object and background combinations.

in images generation (Saharia et al., 2022; Ramesh et al., 2022; Ruiz et al., 2023). As a milestone work, Stable diffusion (Rombach et al., 2022) enables high-fidelity image synthesis with significantly improved efficiency. A branch of work has applied synthetic images in the training of large foundation models (He et al., 2022; Lei et al., 2023; Zhou et al., 2023; Tian et al., 2023). GenImage (Zhu et al., 2023b) proposes a benchmark for the detection of synthetic images from natural ones. To the best of our knowledge, this paper is the first to use synthetic images to expose the vulnerabilities of large foundation models.

## 3 METHODOLOGY

We first present the framework of DiffusionExplorer in Section 3.1, which creates a challenging test set for large foundation models through diffusion generation. We also introduce the statistics and distributions of our test set in Section 3.2.

### 3.1 FRAMEWORK OF DIFFUSIONEXPLORER

Despite the impressive performance of large foundation models, probing these models for their vulnerabilities is important for safety reasons. Typical ways to test models are adopting existing test sets or actively querying the Internet to find failures (Hendrycks et al., 2021). However, images with novel combinations can be missing on the Internet. As shown in Figure 1, an online search for `a plate on the ice skating rink outdoor` on Google does not obtain the expected results, highlighting the absence of such novel image combinations online. While we can conduct real-world experiments or image editing to obtain those images, the collection procedure can be expensive and time-consuming. To address this challenge, we propose to create data with missing combinations using generative models as data sources.

**Generative models as data source for novel image synthesis.** We use the pioneering Stable Diffusion model(Rombach et al., 2022) to obtain images with novel combinations of objects and nuisances, which allows us to create images that follow our text specifications with high fidelity. The generation of Stable Diffusion can be formulated as follows:

$$\text{Image}(C, B) = \text{Stable Diffusion}(\text{Prompt}(C, B)), \tag{1}$$

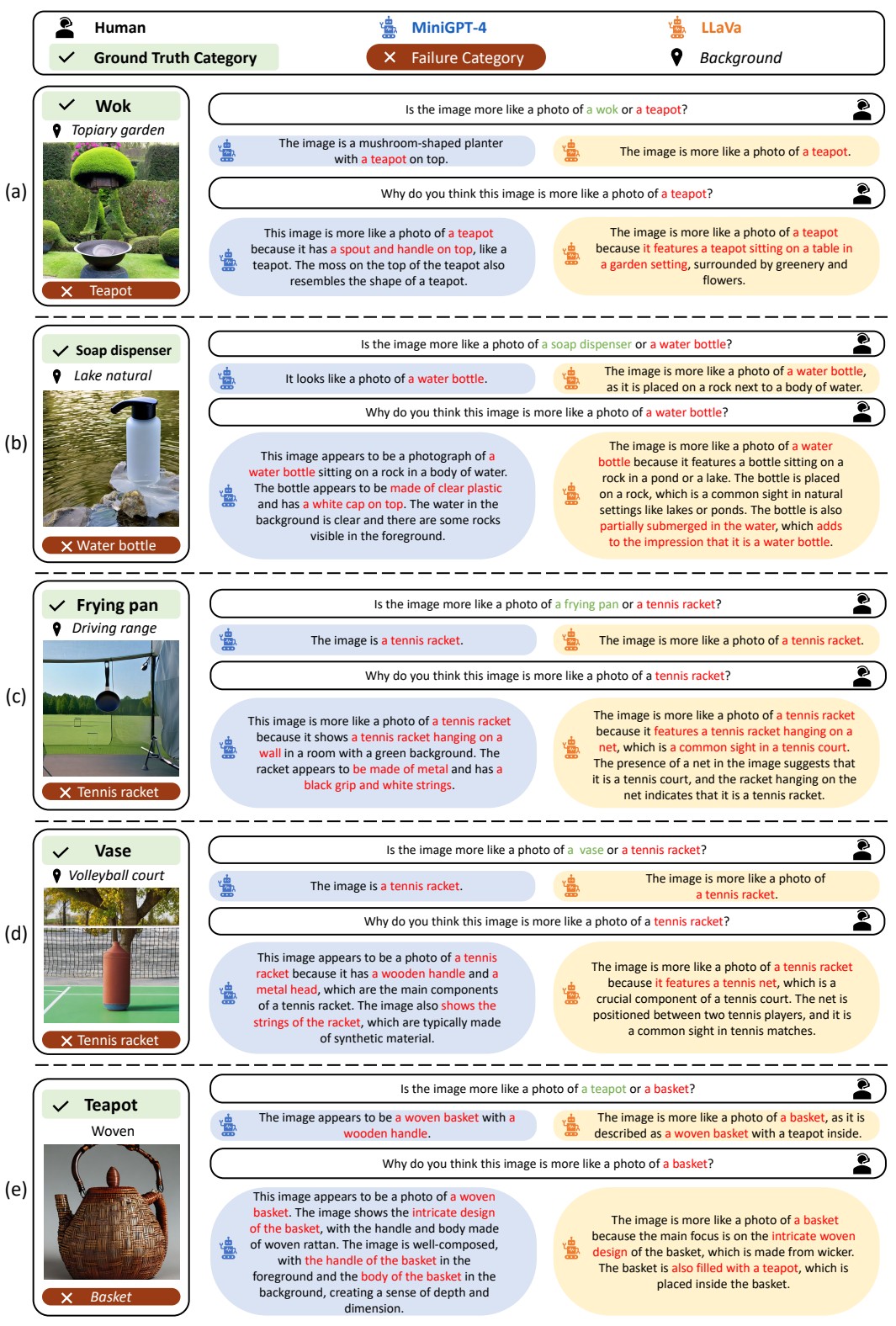

Figure 3: Failure cases of MiniGPT-4 and LLaVa found by DiffusionExplorer. We ask MiniGPT-4 and LLaVa for the category of the input image and the reason for their predictions. For the image (a) to (d), both MiniGPT-4 and LLaVa are fooled by the input images with novel combinations of object and background. For image (e), we create novel combinations of object and texture, e.g., a woven teapot, which can also cause the failure of MiniGPT-4 and LLaVa. Our results show that the generative model can efficiently expose the failures of state-of-the-art foundation models.

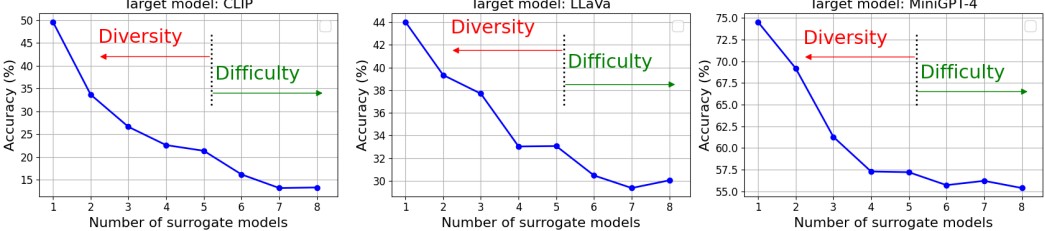

Figure 4: Framework of DiffusionExplorer. DiffusionExplorer first generates images by exhausting all object category and background combinations, then mines the hard test images by finding the shared failure images of surrogate models. For a precise test set, we finalize the test set with human checking. Our approach can find the challenging test images of new large foundation models efficiently.

Figure 5: Test accuracy of target models on shared failures of surrogate models. We adopt known surrogate models to identify their shared failure images as the test set, then evaluate a new target model on this test set. We show that the shared failures of surrogate models can also deceive a new target model, leading to low test accuracy. Increasing the number of surrogate models lowers the target models' test accuracy, suggesting a more difficult test set. Moreover, the test set diversity decreases as more surrogate models are used, leading to a tradeoff between test set diversity and difficulty.

where $C$ and $B$ refer to the object category and background, respectively. Specifically, we present the diffusion model with a $\text{Prompt}(C, B)$ in this format:

$$\texttt{A [category] in the [background]}$$

where we can create images by specifying the object category and background. The versatility of diffusion models allows us to extrapolate and create images with specified category and background, even though this combination is rarely seen in the real-world images.

We first evaluate CLIP (ViT-L/14) on the generated images, which exhaust all object category and background combinations. As shown in Table 1, we find that CILP can correctly classify most of the synthetic images, achieving a high accuracy of 95.29%. To expose the vulnerabilities of large foundation models, we propose an efficient strategy to find the hard test samples among all synthetic images as follows.

Table 1: Test accuracy of CLIP (ViT-L/14). We show that CLIP achieves high accuracy on synthetic images with all object and background pairs.

| Test Set | ImageNet | ObjectNet | Synthetic |
|---|---|---|---|
| Acc (%) | 73.95 | 67.88 | 95.29 |

When creating an image with Stable Diffusion using a specified object category, the vision model should recognize that same category during classification. This helps us detect perception failures in vision models due to category inconsistency. Images deemed hard are those where the model's classification differs from the diffusion input, and these are added to our test set.

**Hard image mining by category consistency.** Since we create an image with a specified object category with Stable Diffusion, the vision model should perceive that same category during classification. This helps us automatically detect perception failures in vision models by category consistency. Specifically, if the predicted category of a classification model differs from the object category in diffusion's input prompt, we view this image to be hard and include it in our test set. However, we find that failure images of a single model can often be correctly classified by another new model, indicated by the high test accuracy with one surrogate model used in Figure 5. We term the unseen new models as *target models* to distinguish the *surrogate models* used in failure image selection. A preferred hard sample should expose the shared vulnerability in multiple vision models, causing concurrent failures. To identify these hard images, we create test sets with images that fail different

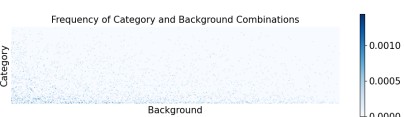

Figure 7: Frequency of object category and background pairs. Each dot indicates a pair, while the x-axis and y-axis indicate the background and category, respectively. A darker color indicates more samples.

numbers of surrogate models, and evaluate them using unseen target models in Figure 5. Figure 5 shows that the test accuracy of target models decreases with more surrogate models used in hard image selection. This trend demonstrates that failure images of multiple surrogate models can form a challenging test set for unseen new models. Notably, increasing the number of surrogate models reduces the diversity of selected images, balancing between test set difficulty and diversity.

**Quality control by human-in-the-loop.** The above algorithm allows us to automatically find a hard test set that can deceive unseen models. However, generative models are not perfect, and they can generate incorrect images that do not contain the specified category in the prompt. For a precise test set, we resort to human annotation to finalize the test set. Figure 2 displays example images from our test set, demonstrating high fidelity and diversity with various object and background combinations. Despite novel combinations, graduate students achieve an estimated 90% accuracy on our test set. The framework of DiffusionExplorer is shown in Figure 4.

## 3.2 DATASET STATISTICS

We generate images using 113 categories from ImageNet and ObjectNet overlap and 468 backgrounds from the Broden dataset (Bau et al., 2017), resulting in a total of 52884 object and background combinations. The lists of category and background are reported in Appendix A and Appendix B, respectively. The final test set after hard sample selection has 5910 images, with an average of 52 images per category, similar to the 50 images per category in ImageNet's test set. Due to the efficiency and flexibility of our framework, more images of new categories and backgrounds can be readily added in our test set. Figure 6

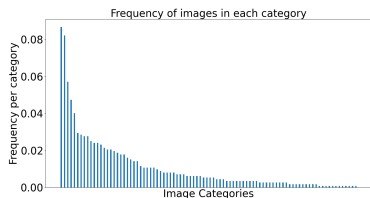

Figure 6: Histogram of the image frequency per category in our test set, following a natural long-tail distribution.

displays the image distribution per category in our test set, showing a natural long-tail pattern. We also show the frequency of different object category and background pairs in Figure 7, indicating a sparse and non-uniform distribution. This non-uniform distribution shows the necessity of exhausting all category and background pairs in test set construction. The frequency of category and background pairs in the lower left corner of Figure 7 is relatively higher than other regions, indicating the higher difficulty of these pairs. Here, we show some example combinations with high frequencies, such as a bench in the badminton court, a paper towel in the corral, and a wok in the forest path. These are all rare object and background combinations in the real world.

## 4 BENCHMARK THE VULNERABILITIES OF FOUNDATION MODELS

We start by describing our experimental setups and present the experimental results of large foundation models on our test set in Section 4.1. We observe that our test set causes a significant accuracy drop of 29.56% and 39.96% for MiniGPT-4 and LLaVa, respectively. We then probe the large foundation models using our test set to uncover more vulnerabilities, including the spurious bias problem and their failures in nearest neighbor retrieval.

**Test set construction setups.** We use Stable Diffusion (Rombach et al., 2022) for image generation due to its high-fidelity synthesis, and use the pretrained weight of version stable-diffusion-2-1 from Hugging Face. As a tradeoff between test set diversity and difficulty, we adopt 4 surrogate models in hard image selection, including CLIP (ViT-L/14), CLIP (ViT-L/14) with resolution 336, CLIP (ResNet50) and a vision model (ResNet50).

**Test set evaluation setups.** For reproducibility, we test open-source vision and foundation models, such as CLIP (Radford et al., 2021), MiniGPT-4 (Zhu et al., 2023a) and LLaVA (Liu et al., 2023). For CLIP, we follow the original paper (Radford et al., 2021) to adopt `A photo of a category` as

Table 2: Test accuracy of vision models and large foundation models. We show the test accuracy for the vision models and large foundation models (rows) on different test sets (columns). The lowest accuracy for each model is **bolded**, indicating a more difficult test set. Our test set achieves the lowest test accuracy for all models. For MiniGPT-4 and LaVa, our test set reduces the accuracy by 29.56% and 39.96% compared to the ImageNet, respectively. Our results show that the generative model can efficiently expose the failures of state-of-the-art foundation models.

| Model | Architecture | ImageNet | ObjectNet | DiffusionExplorer | Resolution |
|---|---|---|---|---|---|
| Vision model (CNN) | VGG11 | 51.90 | 12.84 | **7.80** | 224 |
| | VGG13 | 54.90 | 13.89 | **9.41** | 224 |
| | VGG16 | 58.64 | 17.51 | **13.40** | 224 |
| | VGG19 | 59.30 | 17.95 | **11.87** | 224 |
| | ResNet18 | 54.99 | 14.04 | **8.91** | 224 |
| | ResNet34 | 61.82 | 19.73 | **10.77** | 224 |
| | ResNet101 | 69.62 | 27.91 | **14.50** | 224 |
| | ResNet152 | 69.98 | 29.43 | **15.01** | 224 |
| | Densenet121 | 62.35 | 21.50 | **12.13** | 224 |
| | Densenet161 | 70.99 | 27.62 | **15.35** | 224 |
| | Densenet169 | 68.51 | 25.52 | **15.10** | 224 |
| | Densenet201 | 68.67 | 26.07 | **14.42** | 224 |
| | Wideresnet50 | 73.63 | 29.56 | **10.60** | 224 |
| | Wideresnet101 | 74.11 | 31.74 | **13.99** | 224 |
| Vision model (ViT) | ViT-B/32 | 75.94 | 26.95 | **16.54** | 224 |
| | ViT-B/16 | 82.26 | 36.29 | **20.19** | 224 |
| | ViT-L/16 | 80.03 | 33.69 | **20.36** | 224 |
| CLIP | ResNet101 | 62.00 | 43.60 | **21.15** | 224 |
| | ViT-B/32 | 63.71 | 44.13 | **21.40** | 224 |
| | ViT-B/16 | 67.15 | 55.54 | **22.59** | 224 |
| LLaVa | Vicuna 13B | 73.00 | 66.67 | **33.04** | 336 |
| MiniGPT-4 | Vicuna 13B | 86.84 | 73.91 | **57.28** | 224 |

the text template and report the zero-shot accuracy. MiniGPT-4 and LLaVa provide textual answers to image-based questions. For the evaluation of MiniGPT-4 and LLaVa, we ask the model for the object category and their reasons for predictions with the following questions:

- `Is the image more like a photo of a [Ground truth category] or a [Failure category]?`
- `Why do you think this image is more like a photo of a [Predicted category]?`

We choose the category with the highest CLIP (ViT-L/14) confidence among all incorrect categories as the failure category for the first question. For quantitative evaluation, accuracy is defined as the ratio where the model selects the ground truth category in the response to the first question. Notably, the first question offers two choices, making the random guessing accuracy 50%.

## 4.1 EXPERIMENTAL RESULTS

**Quantitative results.** We report the test accuracy of vision and large foundation models on different test sets in Table 2. Each row is the test accuracy for one model where we compare our test set with ImageNet and ObjectNet. We **bold** the lowest result for each model (row), which indicates the most difficult test set. From Table 2, we can see that our test set achieves the lowest test accuracy for all models, highlighting the difficulty of our test set. For the CLIP model with the architecture ViT-B/16, our test set yields an accuracy of 22.59%, which is 44.56% and 32.95% lower than ImageNet and ObjectNet, respectively. For MiniGPT-4 and LLaVa, our test set reduces the accuracy by 29.56% and 39.96% compared to ImageNet,

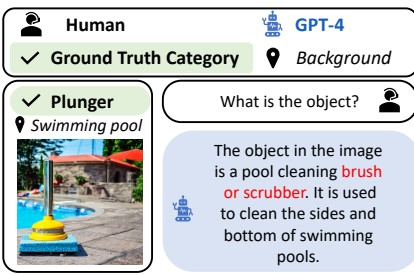

Figure 8: Failure case of GPT-4 in our test set. GPT-4 misclassifies the plunger as a brush or scrubber.

respectively. Notably, the accuracy of MiniGPT-4 and LLaVa is based on the two-choice question introduced in the experimental setups. Therefore, the test accuracy of MiniGPT-4 on our test set

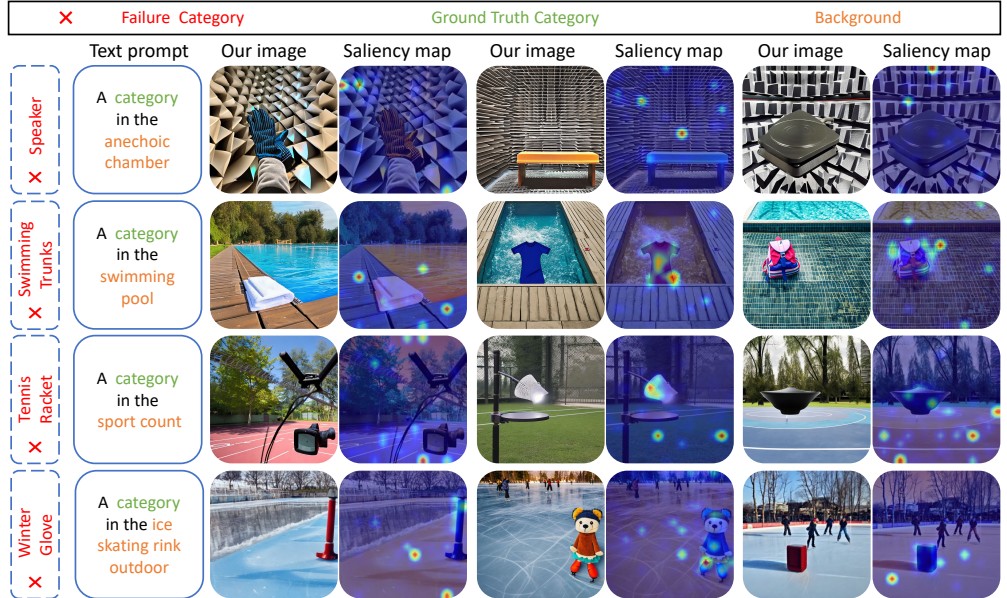

Figure 9: Visualizing the spurious bias in foundation models. For each row, we show examples of images with the same background and different ground truth categories. As shown in the red text of column 1, images in the same row are predicted to the same failure category by CLIP model. Notably, the failure category is highly correlated to the background for each row. In columns 4, 6, and 8, we show the saliency map (Chefer et al., 2021) to see where CLIP model looks when making predictions. Our results reveal that large foundation models can be deceived by spurious cues.

(57.28%) is comparable to random guess accuracy (50%), while the accuracy of LLaVa (33.04%) performs even worse than a random guess. Our results show that the generative model can efficiently expose the failures of state-of-the-art foundation models.

**Visualization results.** We show example conversations with MiniGPT-4 and LLaVa on our test set in Figure 3. Although humans can easily recognize the primary object in images, both MiniGPT-4 and LLaVa mistakenly interpret the image as a failure category. For instance, Figure 3(a) shows a wok in a topiary garden, which is quite a rare object and background combination. Interestingly, MiniGPT-4 understands this image as a photo of the teapot, and explains its reason by claiming that this object *has a spout and handle on top*. We visualize more examples in Appendix C. We also find that our test set is challenging to the latest GPT-4 model and can cause it to mispredict as shown in Figure 8.

## 4.2 CHARACTERIZING THE LARGE FOUNDATION MODELS

To probe large foundation models with our test set for more vulnerabilities, we conduct the following investigations.

**Will large foundation models be deceived by spurious cues?** Previous studies (Geirhos et al., 2020; Arjovsky et al., 2019; Hendrycks et al., 2021) suggest that datasets with spurious cues can cause inflated performance evaluations. For instance, a model might classify an image by its background instead of the object. Our test set, featuring novel object and background combinations, facilitates spurious cue analysis. As shown in Figure 3(c) and Figure 3(d), both MiniGPT-4 and LLaVa misclassify images with a frying pan and vase as a tennis racket due to the background. MiniGPT-4 specifically mentions the tennis net in the background as its reason. We show more examples and visualize their saliency map in Figure 9. Our results highlight that large foundational models can be misled by spurious cues.

**Can CLIP find the correct neighbors of our test images?** CLIP model (Radford et al., 2021) shows potential in the nearest neighbor search tasks. To investigate whether CLIP can find the correct neighbors of our test images, we retrieve the most similar images from ImageNet for our test images. Figure 10 shows that CLIP fails to return the neighbors with the same object category as the query image. As shown in Figure 10(a), retrieved images have a similar background to the query image instead of the object. Moreover, CLIP may retrieve the images that include the object related to the

Figure 10: Visualizations of nearest neighbor images. We visualize the nearest neighbor images with the images from our test set as the query image. Instead of following the same object category as the query image, the nearest neighbor images either follow a similar background or follow another object category that is highly correlated with the background of query image. Our results show that our test set can find the vulnerability of foundation models in nearest neighbor retrieval.

Table 3: Failure transferability of natural images and our synthesized images. We study a new task checking whether examples that cause one model to fail can also lead to the failure of another model, termed failure tranferability. To compare the failure transferability of our synthetic test set and natural test sets, we extract shared failure images of surrogate models from the original ImageNet and ObjectNet, forming new sets: ImageNet (Failure) and ObjectNet (Failure). We show that our test set achieves comparable test accuracy to ImageNet (Failure) and ObjectNet (Failure), indicating that our synthetic images achieve comparable ability to natural images in finding the failures of new models.

| Model | Architecture | ImageNet | ObjectNet | ImageNet (Failure) | ObjectNet (Failure) | DiffusionExplorer |
|-------|-------------|----------|-----------|--------------------|--------------------|------------------|
| CLIP | ResNet101 | 62.00 | 43.60 | 8.85 | 6.85 | 21.15 |
|  | ViT-B/32 | 63.71 | 44.13 | 11.15 | 7.23 | 21.40 |
| LLaVa | Vicuna 13B | 73.00 | 66.67 | 33.27 | 36.86 | 33.04 |
| MiniGPT-4 | Vicuna 13B | 86.84 | 73.91 | 70.46 | 57.06 | 57.28 |

query image's background. In Figure 10(b), a T-shirt retrieves images like pill bottle or chair that are commonly found in an operating room, the background of query image. Our results show that our test set can find the vulnerability of large foundation models in the nearest neighbor search.

**Beyond background: what else can DiffusionExplorer reveal?** Our approach is a flexible framework that can uncover other vulnerabilities in large foundation models beyond background. For instance, while Stylized-ImageNet (Geirhos et al., 2018) exposes texture bias by modifying the styles of ImageNet images, its diversity is limited by the original images. By contrast, DiffusionExplorer can construct a hard test with novel object and texture combinations by specifying the texture in the diffusion models' input prompt. Figure 3(e) shows `a woven teapot` generated by DiffusionExplorer, which both MiniGPT-4 and LLaVa misinterpret as a basket due to the texture. Our results show that DiffusionExplorer can be flexibly extended to expose various vulnerabilities of large foundation models.

**Can our test set match natural ones in failure transferability?** We study a new task checking whether images that cause one model to fail can also lead to the failure of another model, termed *failure transferability*. In Section 3.1, we select hard images by category consistency and show that the shared failures of surrogate models in our test set can also deceive a new target model. This indicates the capability of failure transferability of our synthetic test set. We conduct the same experiment on natural images, including ImageNet (Failure) and ObjectNet (Failure), by finding the shared failure images that all surrogate models misclassify. As shown in Table 3, we find that our generated data achieve similar failure transferability as natural images. In contrast to traditional static datasets like ImageNet and ObjectNet, our method enjoys a lower cost in creating novel data and can be scaled efficiently, suggesting generative models can be a good data source for finding hard test images.

## 5 CONCLUSION

In this paper, we propose a framework that challenges the foundation models by mining hard test samples through diffusion generation. We construct a test set containing various novel combinations of object categories and backgrounds, which shows a general challenge for the large foundation models. Our test set significantly reduces the test accuracy of large foundation models, e.g., an accuracy drop of 29.56% and 39.96% for MiniGPT-4 and LLaVa, respectively. Our work shows that generative models can be a effective data source in finding the vulnerability of large foundation models.

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

## A  CATEGORY OVERVIEW

We generate images using 113 categories from ImageNet and ObjectNet overlap and follow Ob-jectNet (Barbu et al., 2019) for the category names. We present the category list of our test set in alphabetical order as in Table 4. Notably, more object categories can be added to our test set readily due to the efficiency of DiffusionExplorer.

Table 4: Category list of our test set in alphabetical order. There are 113 categories in total, and more categories can be added to our test set readily due to the efficiency of DiffusionExplorer.

| Alarm clock | Backpack | Banana | Band aid | Basket |
|---|---|---|---|---|
| Bath towel | Beer bottle | Bench | Bicycle | Binder (closed) |
| Bottle cap | Bread loaf | Broom | Bucket | Butcher's knife |
| Can opener | Candle | Cellphone | Chair | Clothes hamper |
| Coffee/French press | Combination lock | Computer mouse | Desk lamp | Dishrag or hand towel |
| Doormat | Dress shoe (men) | Drill | Drinking cup | Drying rack for plates |
| Envelope | Fan | Frying pan | Dress | Hair dryer |
| Hammer | Helmet | Iron (for clothes) | Jeans | Keyboard |
| Ladle | Lampshade | Laptop (open) | Lemon | Letter opener |
| Lighter | Lipstick | Matchstick | Measuring cup | Microwave |
| Mixing / Salad bowl | Monitor | Mug | Nail (fastener) | Necklace |
| Orange | Padlock | Paintbrush | Paper towel | Pen |
| Pill bottle | Pillow | Pitcher | Plastic bag | Plate |
| Plunger | Pop can | Portable heater | Printer | Remote control |
| Ruler | Running shoe | Safety pin | Salt shaker | Sandal |
| Screw | Shovel | Skirt | Sleeping bag | Soap dispenser |
| Sock | Soup Bowl | Spatula | Speaker | Still camera |
| Strainer | Stuffed animal | Suit jacket | Sunglasses | Sweater |
| Swimming trunks | T-shirt | TV | Teapot | Tennis racket |
| Tie | Toaster | Toilet paper roll | Trash bin | Tray |
| Umbrella | Vacuum cleaner | Vase | Wallet | Watch |
| Water bottle | Weight (exercise) | Weight scale | Wheel | Whistle |
| Wine bottle | Winter glove | Wok | | |

## B  BACKGROUND OVERVIEW

To generate images with novel object and background combinations, we adopt the 468 backgrounds from the Broden dataset (Bau et al., 2017). We randomly select 100 backgrounds as examples in Table 5.

Table 5: Background examples of our test set in alphabetical order. We show 100 randomly selected examples from all 468 backgrounds for reference. The background list demonstrates the diversity of backgrounds and novel object background combinations in our test set.

| Airplane cabin | Airport | Amphitheater | Apartment building outdoor | Apse indoor |
|---|---|---|---|---|
| Archive | Arrival gate outdoor | Auto factory | Badminton court indoor | Bakery |
| Ballroom | Banquet hall | Barn | Basketball court indoor | Beach |
| Bedroom | Bistro indoor | Bog | Botanical garden | Brewery outdoor |
| Bridge | Bullpen | Bullring | Canal urban | Candy store |
| Canteen | Car interior backseat | Cardroom | Casino indoor | Checkout counter |
| Choir loft exterior | Clothing store | Coast | Coffee shop | Convenience store outdoor |
| Corral | Courtroom | Cubicle library | Diner outdoor | Dinette vehicle |
| Dining hall | Donjon | Doorway indoor | Drainage ditch | Driveway |
| Earth fissure | Elevator freight elevator | Escalator indoor | Fence | Forest broadleaf |
| Formal garden | Fort | Galley | Game room | Granary |
| Gymnasium indoor | Hallway | Hayfield | Highway | Hill |
| Hotel outdoor | Ice cream parlor | Jacuzzi outdoor | Jail indoor | Joss house |
| Junk pile | Kennel outdoor | Kiosk outdoor | Labyrinth indoor | Library indoor |
| Lighthouse | Liquor store indoor | Manufactured home | Marsh | Mission |
| Museum indoor | Natural history museum | Ocean | Parking garage outdoor | Parlor |
| Pasture | Podium indoor | Pub outdoor | Ranch | River |
| Sandbar | Savanna | Shopfront | Shower | Skyscraper |
| Snowfield | Stadium baseball | Street | Subway station corridor | Subway station platform |
| Water mill | Waterfall fan | Wheat field | Wrestling ring indoor | Yard |

## C  MORE FAILURE CASES OF LLAVA AND MINIGPT-4

We show additional examples in our test set that deceive LLaVa and MiniGPT-4 in Figure 11 and Figure 12, respectively. Our results show that DiffusionExplorer can efficiently expose the failures of state-of-the-art foundation models.

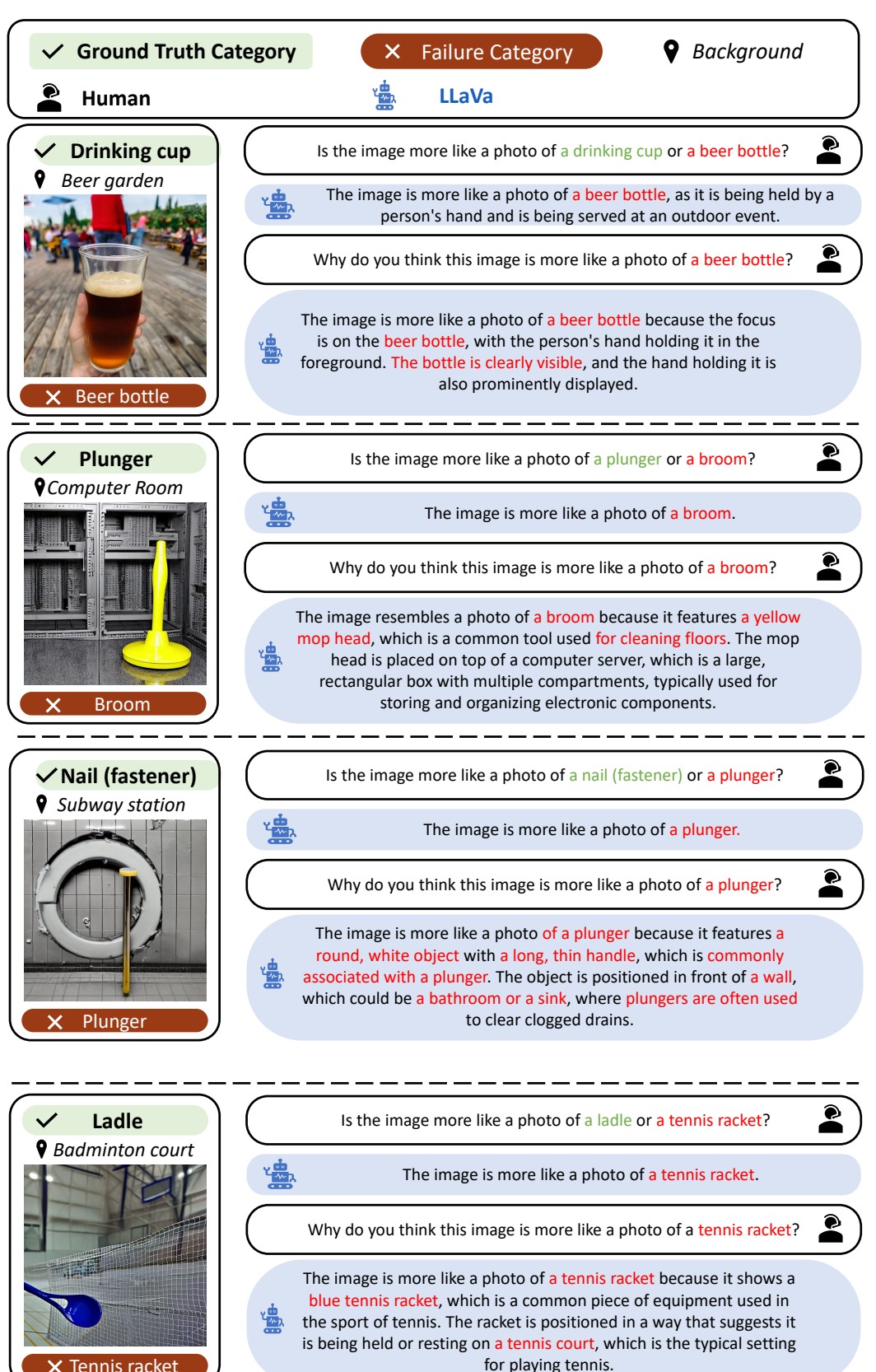

Figure 11: Failure cases of LLaVa found by DiffusionExplorer.

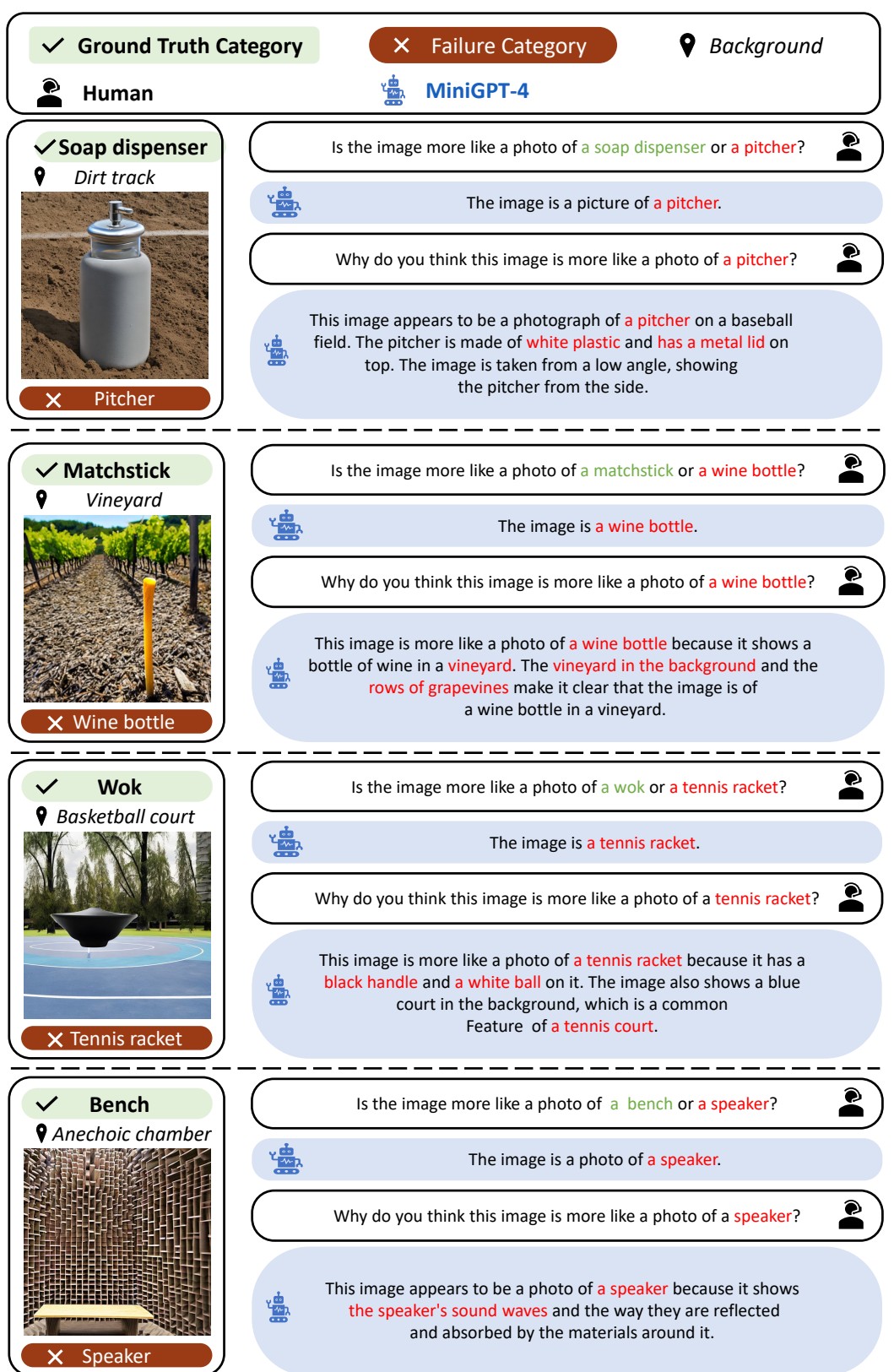

Figure 12: Failure cases of MiniGPT-4 found by DiffusionExplorer.

