# OpenReview forum: "Challenging the Foundations: Mining Hard Test Samples through Diffusion Generation"
_ICLR.cc/2024/Conference — ICLR 2024 Conference Withdrawn Submission_

### Official Review · Reviewer_UyNd · 2023-10-28

**Soundness:** 2 fair
**Presentation:** 3 good
**Contribution:** 2 fair
**Rating:** 3
**Confidence:** 4

**Summary:**

The authors present a framework for creating test datasets that specifically consist of images challenging for foundation models to recognize. This is achieved by employing text-to-image models to generate images featuring uncommon combinations of objects and backgrounds. These generated images are added to the test dataset only if they are misclassified by multiple off-the-shelf vision models. Experimental results demonstrate that well-known foundation models, such as MiniGPT-4 and LLaVa, struggle significantly when faced with these constructed datasets.

**Strengths:**

- The study addresses a critical and currently trending topic in machine learning, focusing on the vulnerability of large-scale foundation models.

- The proposed method effectively generates challenging datasets, leading to a significant decrease in classification accuracy by foundation models.

- The manuscript is well-written and easy to follow.

**Weaknesses:**

- The proposed framework is not specifically designed for applying to foundation models. As it is applicable to any kind of vision model, the motivation of this study is made unclear. In addition, the number of object categories in the created dataset sounds somewhat small, if we consider to evaluate large-scale foundation models.

- The experimental results are not so appealing. It is already well-known that background information can work as spurious cues in vision tasks [R1, R2]. The authors claim that the proposed framework can be also extended to various vulnerabilities "beyond background", but there is no concrete examples or empirical results.

[R1] "Object Recognition with and without Objects," IJCAI 2017.

[R2] "Human Action Recognition without Human," ECCV Workshop, 2016.

- Figure 3 does not provide any essential information on why these models misclassify those images, because the explanations in the answer to the second question is just the output of the model, not coming from the analysis of the inference process by the model such as shown in Figure 9.

- Somewhat related to the above, it is not clear if the misclassification by the foudantion models are really due to the spurious cues incurred by background information. As the text-to-image models generate the images from uncommon combination of objects and backgrounds, the generated images might contain artifacts caused by under-represented prompts, which leads to poor performance of the vision models trained only with natural images.

- How to determine surrogate models is not described clearly.
  - In Figure 8, eight models seem to be used in this experiments, but there is no explanation on what models they are and their orders in use. This experimental result should heavily depend on these points.
  - It seems that the difficulty is measured by the decrese in classification accuracy, but how did the authors measure the diversity?
  - How to choose the four surrogate models used in the experiments at Section 4 is unclear. The authors state that they choose them by considering a tradeoff between test set diversity and difficulty, but detailed information is not provided in the paper, which may doubt the validity of the experimental design.

**Questions:**

Please see weaknesses.

---

### Official Review · Reviewer_NvzL · 2023-11-01

**Soundness:** 3 good
**Presentation:** 3 good
**Contribution:** 2 fair
**Rating:** 5
**Confidence:** 4

**Summary:**

In this paper, the authors proposed DiffusionExplorer which uses diffusion models to generate hard test examples to probe foundation models. Multiple models are evaluated against DiffusionExplorer from CNN models, ViT models, to VLM models and showed great challenges.

**Strengths:**

1. The idea of leveraging diffusion models to generate hard examples is very interesting and the authors also illustrated that some generated images are very confusing for foundation models even thought they are quite easy for human beings.
2. The authors conducted extensive experiments and analysis that showed the examples generated by DiffusionExplorer are indeed quite challenge for a wide spectrum of models.
3. Write is good and easy to follow.

**Weaknesses:**

1.  The images generated by DiffusionExplorer seems quite uncommon and sometime does not make sense in real life. Even though they are quite challenging for foundation models, I am not sure whether these example are a good representation of the performance in real scenarios.
2. The proposed approach seems heavily depends on the performance of Stable Diffusion. As the performance of the diffusion models getting better and better, the images generated by DiffusionExplorer might be less and less challenging.

**Questions:**

1. How valuable are these hard examples given that they are unlike to appear in real applications?

---

### Official Review · Reviewer_YyzH · 2023-11-01

**Soundness:** 2 fair
**Presentation:** 3 good
**Contribution:** 2 fair
**Rating:** 6
**Confidence:** 3

**Summary:**

In this study, the authors provide a new method to evaluate the robustness of foundation models.  The paper highlights the remarkable success of large foundation models in various applications.  The work also points out that these models are often evaluated on typical images, which may lack novel combinations of objects and challenges. To address this, they introduce the "DiffusionExplorer" framework, which efficiently assesses the vulnerabilities of foundation models. Using this framework, they create a test set that exposes the limitations of models like MiniGPT-4 and LLaVa, reducing their accuracy by around 30% and 40%, respectively. The study demonstrates the potential of generative models in revealing the weaknesses of large vision foundation models.

**Strengths:**

- The idea of probing foundation models for their vulnerabilities to the unforeseen category is novel and valuable. The proposed method is aiming at efficiently constructing a test set with novel combinations of object and nuisance factors to expose the failures of foundation models. The could be important to the community.

- The experiment results show a clear drop of accuracy of the existing foundation models, which demonstrate the effectiveness of the proposed method.

**Weaknesses:**

- The motivation is interesting but problematic at the same time. The sample cannot find from Google also means it is not a common case. The test set can be hard but far away from daily life, e.g., anti-physics rules.
- The Prompt(C, B), in a format: A [category] in the [background], seem too simple and cannot provide complex combination. Table 1 is an evidence that the synthetic data are easy when compared with the images from ImageNet.
- The way to choose the hard samples can be further improved: A preferred hard sample should expose the shared vulnerability in multiple vision models, causing concurrent failures.

**Questions:**

- How to evaluate the semantic information of the created images, since they are coming from unseen objects and background combinations?
- Since there are various surrogate models, how to decide which sample is hard, in particular when they are not consistent with each other?
- Should we create the unforseen images or measure the uncertainty of the images? Do we need to generalize the model to the un-exsiting samples? Will this drop the performance to the normal images in our daily life?
- The paper provides a test set with novel combinations of objects and nuisances. can we generate such a set for different models?
Except for the evaluation of the model, can we improve the model?  E.g., Can we use such hard samples to improve the performance on ImageNet by few-shot learning?

---

### Official Review · Reviewer_aTrF · 2023-11-05

**Soundness:** 2 fair
**Presentation:** 3 good
**Contribution:** 1 poor
**Rating:** 3
**Confidence:** 3

**Summary:**

This paper uses diffusion models to construct a new test dataset that is challenging to large models such as LLaVA and MiniGPT4. It is clear that  generative models can be a effective data source for testing foundation models.  Actually, many data synthetic approaches are effective data source for testing large models, including rendering using graphics.

**Strengths:**

1/ A simple method is proposed to mine hard examples for large models using images generated by diffusion models.
2/ A test data set is constructed to challenge the large models including LLaVA and MiniGPT4.

**Weaknesses:**

1/ The approach is very simple. Using generative models to construct  datasets is straightforward. Actually numerous papers have published which use diffusion models to generate training data, not just test data. E.g., DiffuMask: Synthesizing Images with Pixel-level Annotations for Semantic Segmentation Using Diffusion Models. ICCV'23. Note that the task is different. DiffuMask is dealing with semantic segmentation, which is pixel level. However,  for image-level understanding, it's slightly simpler to get the GT label.

2/ Some of the images are even not very clear to human to answer the questions. And in some examples in the paper, the MiniGPT model actually gives sensible answers even if the authors label the answers to be incorrect.

3/ Some cases are more about prompts, instead of confusing the large model. See below my question.

**Questions:**

I tested the last image “Anechoic chamber” at Page 15, by feeding the image to both LLaVA and MiniGPT. The prompt is "Can you describe this image as detailed as possible?". Here are the outputs of both models:
LLaVA: "The image features a room with a large wall made of numerous wooden slats, creating a unique and intricate design. The wall is filled with many small wooden boxes, giving the appearance of a honeycomb structure. In the center of the room, there is a wooden bench, providing a seating area for visitors. The room appears to be a part of a museum or an art installation, as the wooden wall and bench create an interesting and visually appealing environment."

MiniGPT4: "This image shows a room with a wooden bench in the center. The walls are made of wooden planks stacked vertically, creating a grid pattern. The bench appears to be placed in the middle of the room, with the planks forming a backrest for the seat. The room appears to be dimly lit, with no visible light sources. The overall atmosphere of the image is quiet and contemplative."

To me both models outputs make sense.  I do not think both models are confused on this image. The example given in the paper is really about the prompt that is given to the  MiniGPT/LLaVA model "Is the image more like a photo of a bench or a speaker?". This prompt causes confusions, and thus the answer to this prompt is not well defined.  To me both models can understand this content of this image quite well.